# An Elaborate Dynamic Model of the Dual-Motor Precision Transmission Mechanism for Performance Optimization

**Jieji Zheng** [1] , **Xin Xie ***, **Ruoyu Tan, Lingyu Chen** , **Baoyu Li and Dapeng Fan**

College of Intelligence Science and Technology, National University of Defense Technology, Changsha 410073, China
* Correspondence: xiexin12@nudt.edu.cn; Tel.: +86-137-8704-3604

**Abstract:** The dual-motor precision transmission mechanism (DMPTM) is an alternative way to eliminate backlash while ensuring the stiffness of the servo system. However, most of the established models of DMPTM are not accurate enough, and are not conducive to the optimization of system performance and the design of high-precision controllers. In this paper, based on the detailed linear model of the single components of the DMPTM, the dead-zone model, considering the time-varying stiffness, is proposed to describe the backlash of the two transmission chains, and the friction of the mechanism is depicted by the Stribeck model. Then, a high-precision dynamic model of the DMPTM is formed. Finally, the model validation experiments for the open-loop and closed-loop are carried out in the time domain and frequency domain. The experimental results show that the proposed model can accurately describe the nonlinear characteristics of the mechanism. The Pearson correlation coefficient between the proposed model and the actual system is $r_{\text{open-loop}} > 99.41\%$, for the open-loop, and $r_{\text{closed-loop}} > 83.7\%$, for the closed-loop, and these results are both better than those of the existing model. In the frequency domain, whether it is the open-loop or closed-loop model, the frequency response of the proposed model also reproduces the actual system well, which verifies the accuracy of the model.

**Keywords:** dual-motor precision transmission mechanism; detailed linear model; dead zone; friction





## 1. Introduction

Inertially stabilized platforms (ISPs) have been widely used in precision pointing mechanisms, such as remote control weapon stations, tracking radars, optical imaging equipment, antennas, and telescopes [1–4], to isolate the influence of base disturbances on pointing accuracy. As the load of ISPs increases, a torque amplification device must be added to drive the load for high dynamic response. Due to the compact space of the azimuth platform of ISPs, the traditional reducer with a large reduction ratio cannot meet the volume requirements, so the two-stage transmission mode of the planetary reducer and the in series large ring gear (LRG) is still irreplaceable. However, due to the large diameter of the LRG, it is difficult to ensure the manufacturing accuracy of the teeth, resulting in an unpredictable meshing backlash between the gears. Backlash not only deteriorates the control accuracy, reduces the bandwidth, and causes limited cycle oscillation, but also gives rise to nonlinear dynamic responses, such as frequency jump, chaos, and bifurcation [5–9].

For this problem, scholars have proposed various solutions to eliminate backlash. Among them, using the dual-motor precision transmission mechanism (DMPTM) is receiving more and more attention. By applying equal and opposite bias torques to the two sets of motors, it can theoretically eliminate backlash while ensuring the stiffness of the servo system [10–12], which is a distinctive superiority of DMPTM. However, the DMPTM complicates the kinetic properties of ISPs and makes the analysis of the system properties difficult. To improve the control accuracy of the system, an elaborate dynamic model of the DMPTM needs to be established. However, up until now, most of the models of DMPTM

directly simplify the DMPTM to a three-inertia mechanism and apply the traditional dead-zone model to describe the particularity of the backlash [13–15], which is not conducive to the optimization of system performance. The reason is that the traditional dead zone model does not reflect the system backlash characteristics, and the nonlinearity of friction is ignored.

The backlash has a serious impact on system performance, and many mathematical models for backlash and gear play have been published. Ref. [16] reviewed the progress of nonlinear dynamics of gear-driven systems in the past twenty years, especially the gear dynamic behavior considering the backlash. Guesalaga A. [17] established a backlash model using the "bristle" approach. The model introduces a continuous hysteresis representation, different from classical discontinuous models, showing a behavior closer to what has been observed empirically. Merzouki R. [18] modeled disturbing backlash torque by a continuous and derivable mathematical function describing the opposite of the sigmoid function. Barbosa RS [19] and Duarte FB [20] analyzed the dynamical properties of systems with backlash and impact phenomena based on the describing function method. However, the describing function has the defect that it can only be used for frequency domain stability analysis. The dead-zone model [21] is the most widely used to depict backlash in the time domain. Due to the non-differentiable property of the traditional dead-zone model, which makes the control design problem very complex and difficult, Shi Z [22] proposes a "soft degree" concept based on a recently developed differentiable dead-zone model and then presents a practical backstepping algorithm to achieve not only high-precision output tracing, but also limited cycle elimination. Yongjun S. [23] considered the time-varying stiffness based on the dead-zone model when studying the nonlinear dynamics of the gear pair based on the incremental harmonic balance method. Based on the results presented in his paper, the periodic solution with arbitrary precision can be expeditiously obtained, which is useful in analyzing or controlling the dynamics of the gear system. A shortcoming of the studies above is that the non-contact area is regarded as having no output by ignoring the influence of viscous damping. Aiming at this phenomenon, Kranawetter K. [24] proposed a new dead-zone model based on phase plane analysis, which takes into account the damping of gears in the non-contact phase and corrects the boundary of the non-contact area. The accuracy of the models is improved, providing a reference for the backlash analysis in this paper, but these models do not consider the friction nonlinearity of the gears.

In summary, the existing research on the accurate modeling of the transmission chain in the field of the DMPTM has not yet generated relevant reports. When studying the nonlinearity of backlash, friction is not given enough attention, resulting in a comparatively large deviation from the actual system.

Compared with previous studies, this paper contributes the following two works: (1) A case study of the complex transmission chain of the DMPTM is first worked out in detail; (2) The time-varying stiffness is premeditated in the modified dead-zone model.

The organization of this study is as follows. The detailed linear model of the transmission chain of the DMPTM is established in Section 2., while the dead-zone model considering the time-varying stiffness and the friction model of the system are established in Section 3. Then, the accuracy of the proposed model is experimentally verified in the time domain and the frequency domain in Section 4. Finally, the conclusion of this study is summarized in Section 5.

## 2. Component Modeling

### 2.1. Overall Structure of the DMPTM

The system presented in this paper is shown in Figure 1. The structure of a typical DMPTM is sketched in Figure 1, which consists of two identical transmission chains (including a permanent magnet synchronous motor (PMSM), an L-shaped planetary reducer (LSPR), a pinion, and an LRG.

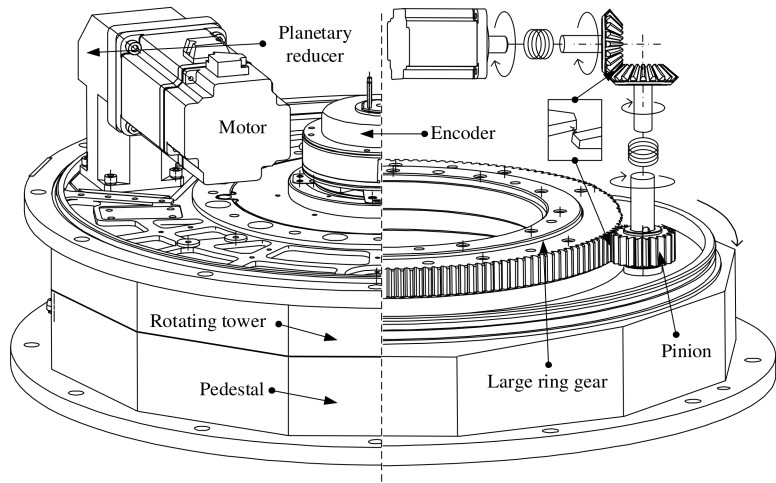

**Figure 1.** Schematic diagram of the transmission structure of the DMPTM.

First of all, a detailed linear model of the DMPTM has been derived, which reveals the torque transmission principle of the mechanism. In this process, it is necessary to identify the degrees of freedom that mainly contribute to the exchange of mechanical actions. The degrees of freedom included in the model are those that are directly implied by the transmission of the torque from the motor to the last stage of the transmission. Details on modeling the single components are reported in the following subsections.

### 2.2. Motor and Driving Wheel of LSPR

Figure 2 shows the coupling between the motor and the driving wheel of the LSPR, whose equations are easily derived as a two-mass compliant system:

$$J_m\ddot{\theta}_0 + D_m\dot{\theta}_0 = T_m - T_{01} \tag{1}$$

$$J_1\ddot{\theta}_1 + D_1\dot{\theta}_1 = T_{01} - F_{12t}R_1 \tag{2}$$

$$T_m = K_dK_Mu \tag{3}$$

$$T_{01} = K_{01}\Delta\theta_{01} \tag{4}$$

$$\Delta\theta_{01} = \theta_0 - \theta_1 \tag{5}$$

where $J_m, D_m$ are the inertia and damping of the motor rotor, while $J_1, D_1$ are the inertia and damping of the driving wheel of the LSPR; $\theta_0, \theta_1$ are the rotation angle of the motor rotor and the driving wheel, respectively; $K_{01}$ is the stiffness of the motor shaft; $T_m, T_{01}$ are the electromagnetic torque of the motor and the input torque of the LSPR, respectively; $K_d$ is the amplification factor of the driver, while $K_M$ is the torque constant of the motor; $u$ is the input voltage of the motor; $F_{12t}$ is the circumferential force of the driving wheel; and $R_1$ is the radius of the driving wheel.

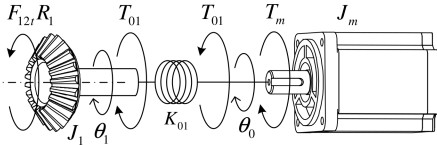

**Figure 2.** Motor and the driving wheel of the LSPR coupling.

### 2.3. Internal Bevel Gear Drive of LSPR

The LSPR is composed of a planetary gear stage and a bevel gear stage. As the first stage, the backlash of the planetary stage is so small that it can be ignored, and its friction can be equivalent to the motor side. Therefore, the focus is on the bevel gear transmission.

Figure 3 shows the bevel gear transmission inside the reducer. The dynamic equation of the LSPR is:

$$J_2\ddot{\theta}_2 + D_2\dot{\theta}_2 = F_{12t} \cdot R_2 - T_{23} \tag{6}$$

where $J_2, D_2$ are the inertia and damping of the driven wheel of the LSPR; $R_2$ is the radius of the driven wheel; and $T_{23}$ is the output torque of the LSPR. The force analysis of the bevel gear teeth is shown in Figure 4, and the circumferential force $F_{12t}$ and normal force $F_{12n}$ when the gear teeth mesh can be obtained as follows:

$$F_{12t} = F_{12n}\cos\alpha_1 \tag{7}$$

$$F_{12n} = K_{12}\Delta L_{12} + D_{12}\Delta\dot{L}_{12} \tag{8}$$

where $\alpha_1$ is the pressure angle of the bevel gear, $K_{12}, D_{12}$ are the meshing stiffness and meshing damping of the bevel gear, respectively; $\Delta L_{12}$ is the deformation of the teeth of the bevel gear. Figure 5 shows the relative motion of gear meshing, and the deformation $\Delta L_{12}$ can be expressed by Equation (9):

$$\Delta L_{12} = (\theta_1 R_1 - \theta_2 R_2)\cos\alpha_1\cos\delta_1 \tag{9}$$

where $\delta_1$ is the taper angle of the bevel gear.

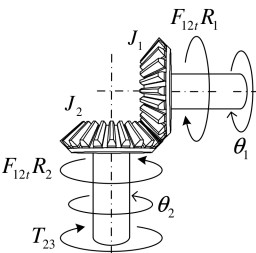

**Figure 3.** Bevel gear meshing transmission of the LSPR.

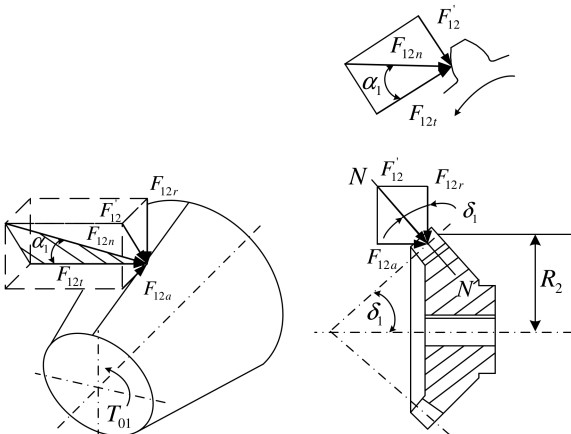

**Figure 4.** Force analysis of the bevel gear teeth.

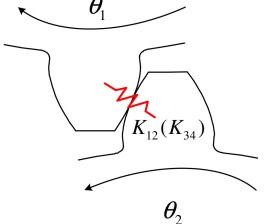

**Figure 5.** The relative motion of gear meshing.

### 2.4. Driven Wheel of LSPR and Pinion

The coupling of the driven wheel of the LSPR to the pinion is shown in Figure 6, and the dynamic equation can be described as:

$$J_3\ddot{\theta}_3 + D_3\dot{\theta}_3 = T_{23} - F_{34t}R_3 \tag{10}$$

$$T_{23} = K_{23}\Delta\theta_{23} \tag{11}$$

$$\Delta\theta_{23} = \theta_2 - \theta_3 \tag{12}$$

$$F_{34t} = (K_{34}\Delta L_{34} + D_{34}\Delta\dot{L}_{34})\cos\alpha_3 \tag{13}$$

where $J_3, D_3$ are the inertia and damping of the pinion, while $K_{23}$ is the stiffness of the output shaft; $\theta_2, \theta_3$ are the rotation angle of the driven wheel and the pinion, respectively; $F_{34t}$ is the circumferential force of the pinion, and $R_3$ is the radius of the pinion; $K_{34}, D_{34}$ are the meshing stiffness and meshing damping of the pinion, respectively; $\alpha_3$ is the pressure angle of the gear, and $\Delta L_{34}$ is the deformation of the teeth of the gears. The deformation $\Delta L_{34}$ can be expressed by Equation (14):

$$\Delta L_{34} = (\theta_3 R_3 - \theta_L R_L)\cos\alpha_3 \tag{14}$$

where $\theta_L, R_L$ are the rotation angle and the radius of the LRG, respectively.

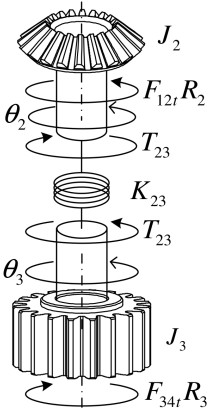

**Figure 6.** Coupling between the driven wheel of the LSPR and the pinion.

### 2.5. Pinions and LRG

Figure 7 shows a schematic diagram of the coupling between one of the pinions and the LGR, and Figure 8 shows the meshing dynamics model of the two pinions and the LGR. By analyzing Figures 7 and 8, the dynamic equation of the LGR can be obtained as:

$$\begin{aligned}
J_L\ddot{\theta}_L + D_L\dot{\theta}_L &= F_{l34t}R_L + F_{r34t}R_L \\
&= (K_{34}\Delta L_{l34} + D_{34}\Delta\dot{L}_{l34})R_L\cos\alpha_3 \\
&\quad + (K_{34}\Delta L_{r34} + D_{34}\Delta\dot{L}_{r34})R_L\cos\alpha_3
\end{aligned} \tag{15}$$

where $J_L, D_L$ are the inertia and damping of the LRG; $F_{l34t}, F_{r34t}$ are the circumferential forces of the LGR on both sides; $\Delta L_{l34}, \Delta L_{r34}$ are the deformations of the teeth of the gears.

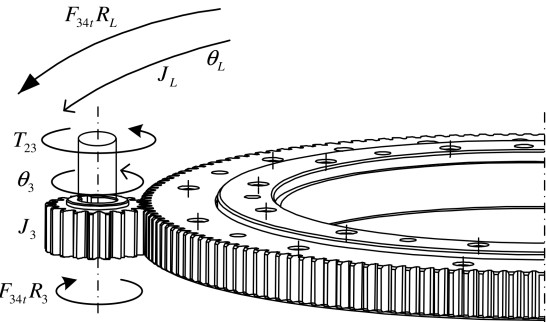

**Figure 7.** Pinion and LGR coupling.

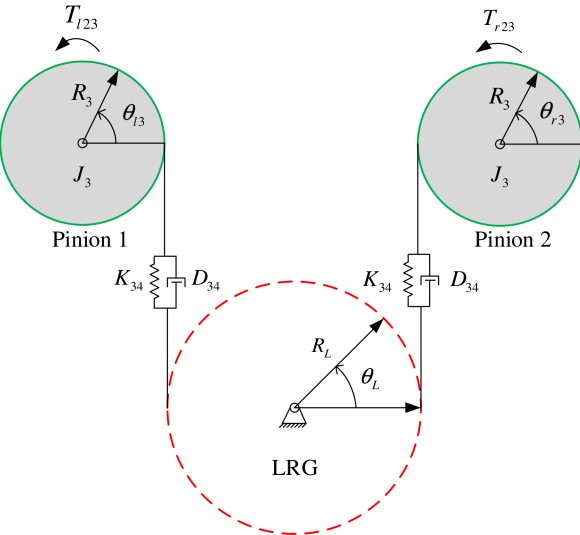

**Figure 8.** The dynamic model of meshing two pinions with the LGR.

### 2.6. Overall Linear Model

The equations derived above are all linear, forming a detailed linear model of the overall transmission chain of the DMPTM, which can be represented by Figure 9.

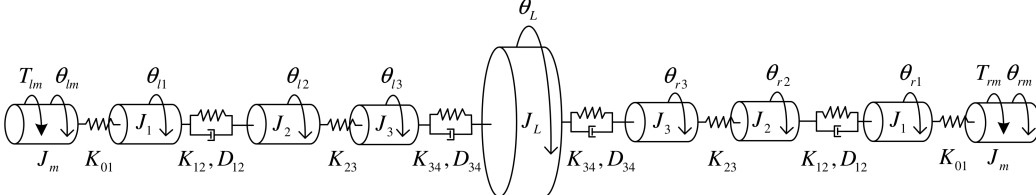

**Figure 9.** The overall linear model of the DMPTM.

The model depicted in Figure 9 contains 9 degrees of freedom, resulting in a very complex model. To analyze the factors that have a greater impact on the system performance inside the system, we can find ways to reasonably simplify the model. The rigidity of the motor shaft is large enough and the backlash of the LSPRs is within 7 arcmin. Moreover, the internal detailed parameters of the LSPRs are not convenient to obtain, since they form the proprietary background of the manufacturer of the transmission. Hence, the motors and the LSPRs can be regarded as ideal transmission links, so $\theta_{lm} \approx \theta_{l1}$, $\theta_{rm} \approx \theta_{r1}$, $\theta_{l2} \approx \theta_{l3}$, $\theta_{r2} \approx \theta_{r3}$ are obtained. The relationship of the motor angle to the pinion angle can be simplified as:

$$\theta_{lm} \approx N_1 \theta_{l3}, \theta_{rm} \approx N_1 \theta_{r3} \tag{16}$$

where $N_1$ is the reduction ratio of the LSPR. The simplified overall linear model of the system is shown in Figure 10.

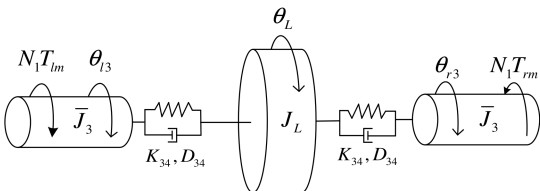

**Figure 10.** The simplified overall linear model of the DMPTM.

Where $\bar{J}_3 = N_1^2 J_m + J_1 + J_2 + J_3$ in the figure is the sum of the inertia equivalent to the pinion end. The simplified system vibration equation is shown in Equation (17):

$$\overline{M}\ddot{\overline{X}} + \overline{D}\dot{\overline{X}} + \overline{K}\overline{X} = \overline{E}T \tag{17}$$

where

$$T = \begin{bmatrix} T_{lm} & T_{rm} & 0 \end{bmatrix}^T$$

$$\overline{E} = \begin{bmatrix} N_1 & 0 & 0 \\ 0 & N_1 & 0 \\ 0 & 0 & 0 \end{bmatrix}$$

$$\overline{X} = \begin{bmatrix} \theta_{l3} & \theta_{r3} & \theta_L \end{bmatrix}^T$$

$$\overline{M} = diag\begin{bmatrix} \bar{J}_3 & \bar{J}_3 & J_L \end{bmatrix}$$

See Appendix A for the expression of matrix $\overline{D}$ and $\overline{K}$. Neglecting the effect of damping in the vibration equation, Equation (17) can be simplified to an undamped free vibration equation:

$$\overline{M}\ddot{\overline{X}} + \overline{K}\overline{X} = \overline{E}T \tag{18}$$

Therefore, the resonant frequency $\omega_{NTF}$ of the system satisfies:

$$\overline{K} - \omega_{NTF}^2 \overline{M} = 0 \tag{19}$$

Substitute $\overline{M}$ and $\overline{K}$ into Formula (19) to get the determinant:

$$\det \begin{vmatrix} \bar{k}_{1,1} - \omega_{NTF}^2 \bar{J}_3 & 0 & \bar{k}_{1,3} \\ 0 & \bar{k}_{2,2} - \omega_{NTF}^2 \bar{J}_3 & \bar{k}_{2,3} \\ \bar{k}_{3,1} & \bar{k}_{3,2} & \bar{k}_{3,3} - \omega_{NTF}^2 J_L \end{vmatrix} = 0 \tag{20}$$

The system resonance frequency can be obtained by solving Equation (20).

## 3. Analysis of Nonlinear Dynamics

The two transmission chains of the DMPTM consist of two stages. The first stage is an LSPR, whose backlash on the system after deceleration can be ignored, but its friction is nonnegligible. The second stage consists of a pinion and an LRG, whose manufacturing and assembly errors make the backlash large, which directly affects the servo performance of the system. At the same time, the friction of the LRG is also a key factor affecting the performance.

Based on the detailed linear model established in Section 2, a modified dead-zone model considering the time-varying stiffness is proposed to depict the backlash between the pinion and the LRG. Furthermore, the Stribeck model is adopted to describe the friction.

### 3.1. Modified Dead-Zone Model

The backlash of gears is shown in Figure 11, and $2\Delta$ represents the backlash. According to Ref. [20], the dead zone model of the pinion gear and the LRG can be expressed by Equation (21):

$$\tau_c = \begin{cases} k(z - \Delta) + c\dot{z} & (z, \dot{z}) \in A^+ \\ c_f \dot{z} & (z, \dot{z}) \in A_0 \\ k(z + \Delta) + c\dot{z} & (z, \dot{z}) \in A^- \end{cases} \tag{21}$$

where

$$A^+ = \left\{ (z, \dot{z}) : \right.$$
$$\left\{ \begin{array}{ll} z - \frac{c_f}{k}\dot{z} + \frac{c+c_f}{k}\dot{z}e^{-\frac{k}{c+c_f}\left(\frac{z+b}{\dot{z}} + \frac{c}{k}\right)} \geq \Delta, \dot{z} > 0 \\ k(z - \Delta) + c\dot{z} \geq 0, & \forall \dot{z} \end{array} \right\}$$
$$A^- = \left\{ (z, \dot{z}) : \right.$$
$$\left\{ \begin{array}{ll} z - \frac{c_f}{k}\dot{z} + \frac{c+c_f}{k}\dot{z}e^{-\frac{k}{c+c_f}\left(\frac{z-b}{\dot{z}} + \frac{c}{k}\right)} \leq -\Delta, \dot{z} < 0 \\ k(z - \Delta) + \dot{z} \leq 0, & \forall \dot{z} \end{array} \right\}$$
$$A_0 = \left\{ (z, \dot{z}) \right\} \backslash (A^+ \cup A^-)$$

where $k$ and $c$ are the meshing stiffness and damping, respectively; $z = \theta_3 - N_2\theta_L = \theta_3 - \frac{z_L}{z_3}\theta_L$ is the transmission error between the pinion and the LRG, while $z_3, z_L$ are the number of teeth of the pinion and the LRG, respectively; $c_f$ is the damping when the pinion and the LRG are in a non-contact state.

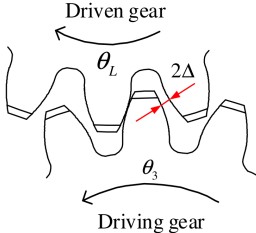

**Figure 11.** Backlash of gears.

The dead-zone model in Ref. [20] ignored the time-varying characteristics of stiffness, resulting in the inaccuracy of the model. The stiffness $k$ in the teeth engagements (between the pinion gear and the LRG) has been computed, referring to a simplified scheme of the tooth, sketched in Figure 12. The tooth is represented as a clamped bracket with variable sections and with the force applied to the pitch circle of the wheel. The stiffness of the tooth has been computed with the formula:

$$k = \left( \int_0^{h_1} \frac{x^2}{EJ(x)} dx \right)^{-1} \tag{22}$$

where $h_1$ is the distance from the base of the tooth to the pitch circle of the tooth; $E$ is Young's modulus of gears; $x$ is the linear coordinate along the distance, and $J(x)$ is the polar moment of inertia. For the specific calculation process of $J(x)$, please refer to Ref. [25].

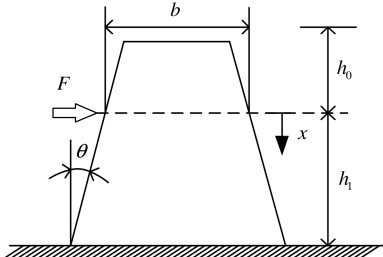

**Figure 12.** Scheme of the tooth profile.

Convert $k$ to the stiffness in the circumferential direction of the gear as the gear meshing average stiffness $k_m$. The time-varying mesh stiffness $k(t)$ of gear meshing can be expressed by Equation (23) [21]:

$$k(t) = k_m + k_a \cos(\omega_z t + \varphi_r) \tag{23}$$

where $k_a$ is the amplitude of the stiffness variation; $\omega_z$ is the fundamental frequency of the stiffness, and $\omega_z = 2\pi n_3 z_3 / 60$; $n_3$ is the pinion rotary speed; $\varphi_r$ is the phase angle of the stiffness, and $\varphi_r = 0$.

The meshing damping of gears can be expressed by Equation (24) [21]:

$$c = 2\zeta \sqrt{\frac{k_m R_3^2 R_L^2 J_3 J_L}{R_3^2 J_3 + R_L^2 J_L}} \tag{24}$$

where $\zeta$ is the damping ratio, which is taken as 0.03–0.17.

### 3.2. Stribeck Friction Model

The frictions of the system are all represented by the Stribeck model, and its expression is as follows [26]:

$$\mathbf{T}_f\left(\dot{\theta}, \mathbf{T}_m\right) = \begin{cases} \mathbf{T}_m, \text{ if } \left(\dot{\theta} = 0 \text{ and } \mathbf{T}_s^- < \mathbf{T}_m < \mathbf{T}_s^+\right) \\ \mathbf{T}_s^+, \text{ if } \left(\dot{\theta} = 0 \text{ and } \mathbf{T}_m > \mathbf{T}_s^+\right) \\ \mathbf{T}_s^-, \text{ if } \left(\dot{\theta} = 0 \text{ and } \mathbf{T}_m < \mathbf{T}_s^-\right) \\ \mathbf{T}_C^+ + \left(\mathbf{T}_s^+ - \mathbf{T}_c^+\right) e^{-(\dot{\theta}/\Omega_+)} + B^+ \dot{\theta}, \text{ if } \left(\dot{\theta} > 0\right) \\ \mathbf{T}_C^- + \left(\mathbf{T}_s^- - \mathbf{T}_c^-\right) e^{-(\dot{\theta}/\Omega_-)} + B^- \dot{\theta}, \text{ if } \left(\dot{\theta} < 0\right) \end{cases} \tag{25}$$

where $\mathbf{T}_f$ is the friction torque; $\mathbf{T}_s^+$, $\mathbf{T}_s^-$ are the static friction torques; $\mathbf{T}_c^+$, $\mathbf{T}_c^-$ are the Coulomb friction torques; $\Omega_+$, $\Omega_-$ are the Stribeck velocities; and $B^+$, $B^-$ are the viscous dampings.

### 3.3. Overall Dynamic Model

First, the linear model of the DMPTM transmission chain is established step by step, according to the torque transmission process, and then the modified dead-zone model, considering time-varying stiffness and the Stribeck friction model, are added. The overall dynamic model of DMPTM finally established is shown in Figure 13. In the figure, $T_{f1}, T_{f2}, T_{fL}$ are the friction of the two LSPRs and the LRG, while $\Delta_1, \Delta_2$ are the backlash between the two pinions and the LRG, respectively. In the actual system, due to the uncertainty of manufacturing and assembly, the backlash and friction of the two transmission chains are different. Therefore, the following model parameters need to be obtained through separate identification.

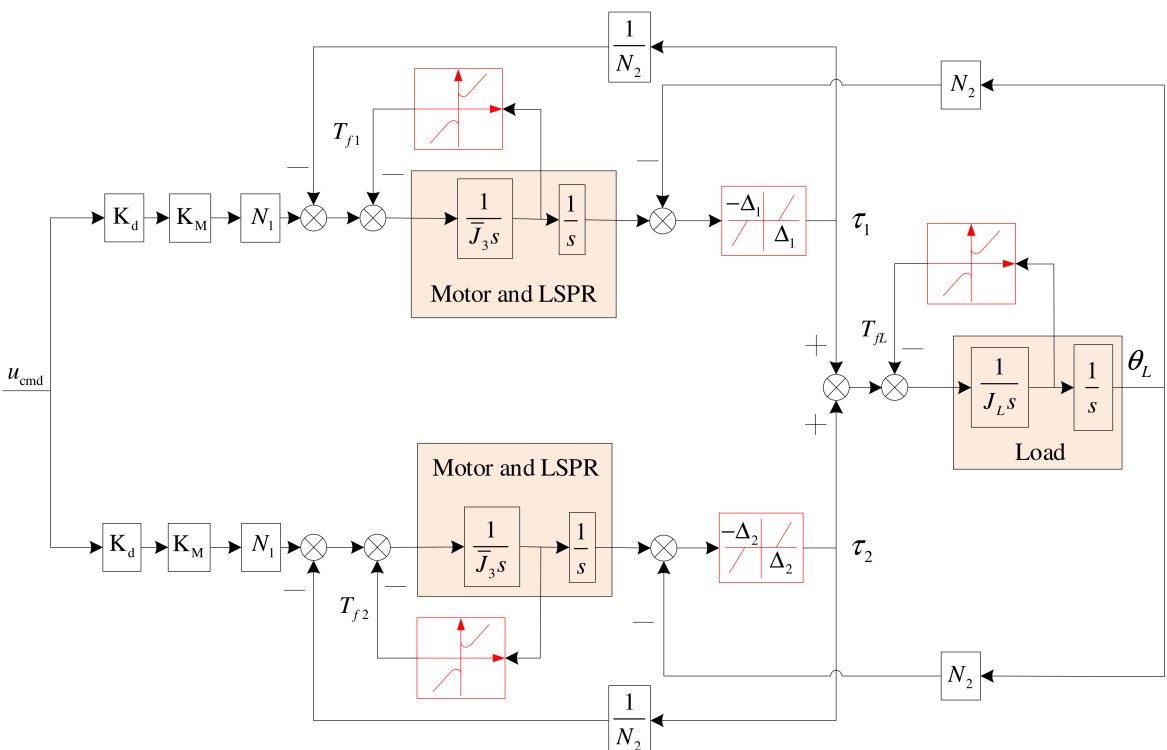

**Figure 13.** Block diagram of the nonlinear dynamic model of the DMPTM.

## 4. Model Validation

### 4.1. Experimental Setup

A DMPTM experimental device, as shown in Figure 14, was built, mainly consisting of two PMSMs (model: SPALY80), two drivers (model: Elmo P/N: SOL-WHI 20/100PYE), two LSPRs (model: FABR060-25-S2-P1), an azimuth platform, an absolute encoder (model: CAPRO-B112050), a fiber optic gyroscope (model: FOG-118), a 24 V power supply, a 48 V power supply, dSPACE1104, and an industrial computer.

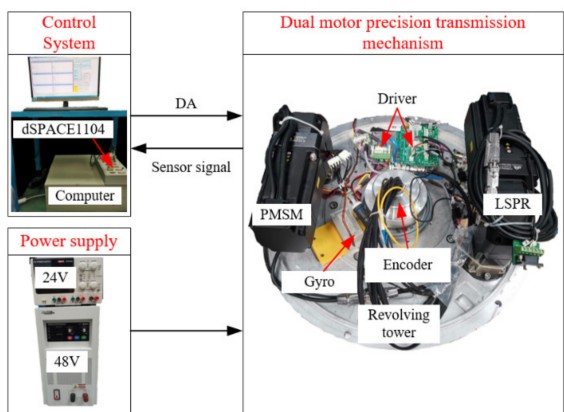

**Figure 14.** Experimental device for testing the DMPT.

### 4.2. Determination of Model Parameters

The kinetic model worked out so far involves many mechanical parameters, whose accurate knowledge is essential. An analysis has been performed to derive expressions for these parameters in terms of the geometrical properties of the bodies, and the properties (inertial and elastic) of the materials. This analysis is briefly summarized hereafter.

### 4.2.1. Inertia Parameter

The acquisition of the inertia parameters is relatively simple. The inertia of the motor and reducer is directly provided by the manufacturer, and the inertia of the pinion gear, the LRG, and the rotating tower can be calculated using a 3D modeler.

### 4.2.2. Dead-Zone Model Parameters

The parameters that need to be obtained for the dead-zone model mainly include the size $\Delta_1, \Delta_2$ of the meshing backlash between the two pinions and the LRG teeth and the dead zone transition damping $c_f$.

The value of $\Delta_1, \Delta_2$ is mainly obtained through the experimental test. In this paper, the reverse motion measurement method is used to obtain the $\Delta_1, \Delta_2$ through the difference between the forward and reverse rotation angles. Dead-zone transition damping $c_f$ is an adjustable parameter, and it takes a value with a higher degree of fitting with the measured data.

### 4.2.3. Friction Model Parameters

The Stribeck friction models of the LSPRs and the LRG have been obtained through standard identification experiments and are not reported here for brevity.

By consulting the parameters provided by the equipment manufacturer and the test results, the system parameter values shown in Table 1 are obtained. Table 2 shows the friction model parameters of the LSPRs and the LRG.

**Table 1.** System parameter values.

| Symbol | Value | Symbol | Value |
|---|---|---|---|
| $K_d$ (A·V$^{-1}$) | 3 | $z_3$ | 20 |
| $K_M$ (Nm·A$^{-1}$) | 0.13 | $z_L$ | 165 |
| $J_m$ (kg·m$^2$) | $2 \times 10^{-4}$ | $k_m$ (Nm·rad$^{-1}$) | $3.6 \times 10^4$ |
| $J_2$ (kg·m$^2$) | 0.125 | $c$ (Nm·deg$^{-1}$·s) | 1 |
| $J_L$ (kg·m$^2$) | 2.5 | $\Delta_1$ (arcmin) | 14 |
| $N_1$ | 25 | $\Delta_2$ (arcmin) | 10 |
| $N_2$ | 8.25 | | |

**Table 2.** Friction model parameter values.

| Friction of the LSPRs | | Friction of the LRG | |
|---|---|---|---|
| Symbol | Value | Symbol | Value |
| $\mathbf{T}_{3s}^+$ (Nm) | 2.5 | $\mathbf{T}_{Ls}^+$ (Nm) | 23 |
| $\mathbf{T}_{3s}^-$ (Nm) | $-2$ | $\mathbf{T}_{Ls}^-$ (Nm) | $-20$ |
| $\mathbf{T}_{3c}^+$ (Nm) | 2.8 | $\mathbf{T}_{Lc}^+$ (Nm) | 20 |
| $\mathbf{T}_{3c}^-$ (Nm) | $-1.8$ | $\mathbf{T}_{Lc}^-$ (Nm) | $-17$ |
| $\Omega_{3+}$ (°/s) | 2 | $\Omega_{L+}$ (Nm) | 1 |
| $\Omega_{3-}$ (Nm) | $-2$ | $\Omega_{L-}$ (Nm) | $-1$ |
| $B_3^+$ (Nm·deg$^{-1}$·s) | 0.01 | $B_L^+$ (Nm·deg$^{-1}$·s) | 1.2 |
| $B_3^-$ (Nm·deg$^{-1}$·s) | 0.008 | $B_L^-$ (Nm·deg$^{-1}$·s) | 1 |

### *4.3. Experimental Results*

To fully verify the accuracy of the model, the model response and the actual system response when the system is open-looped and closed-looped are compared in the time domain and frequency domain, respectively. Reference would be made to the velocity control loop (Figure 15); the loop is closed on the load side, the speed measurements being numerical differentiations of position measurements obtained with an encoder. A PI (proportional integral) regulator $G_c(s) = k_p + k_i \frac{1}{s}$ is used, whose tuning had already been performed independently of this analysis. This tuning, however, is essential here, as the goal is just the validation of the model.

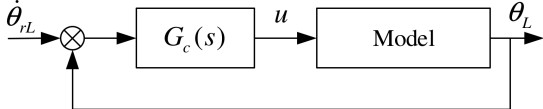

**Figure 15.** PI speed control system.

In the figure, $\dot{\theta}_{rL}$ is the velocity setpoint, and $\dot{\theta}_L$ is the velocity response of the LRG.

### 4.3.1. Open-Loop Time Response

First, the open-loop time-domain responses of the system are verified experimentally. Different excitation signals are applied to the motor, and the comparison results are shown in Figure 16a–d. In the figure, the red solid lines are the model established according to the existing method of Ref. [10], the blue solid lines are the model proposed in this paper, and the black dotted lines are the response of the actual system. It can be seen from the comparison results that the fitting degrees of the proposed model and the actual system are higher than that of the existing model, under either the excitation of sinusoidal signals or square wave signals, with different amplitudes and frequencies. The proposed model can better reproduce the zero-crossing dead-zone characteristics and the transition time of the commutation of the actual system, which are the key factors affecting the performance of the servo system. It can be found from Figure 16 that the DMPTM has a large dead zone and jitter in the process of changing direction due to the existence of backlash and friction.

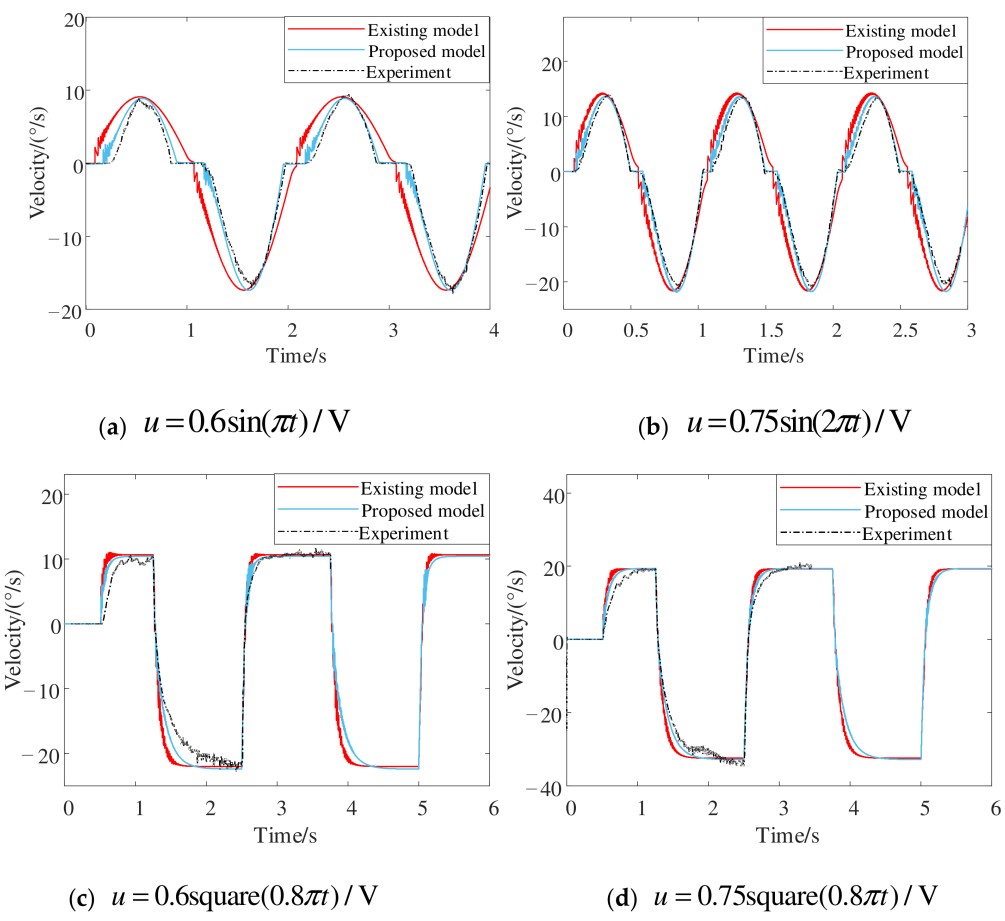

**Figure 16.** Simulated and experimental time responses under an open-loop.

### 4.3.2. Open-Loop Frequency Response

In the frequency domain, the frequency responses of the model and the system at different voltage amplitudes are also contrasted, and the results are shown in Figure 17a,b. The results show that the amplitude and phase of the proposed model also fit the system frequency response characteristics better than the existing model.

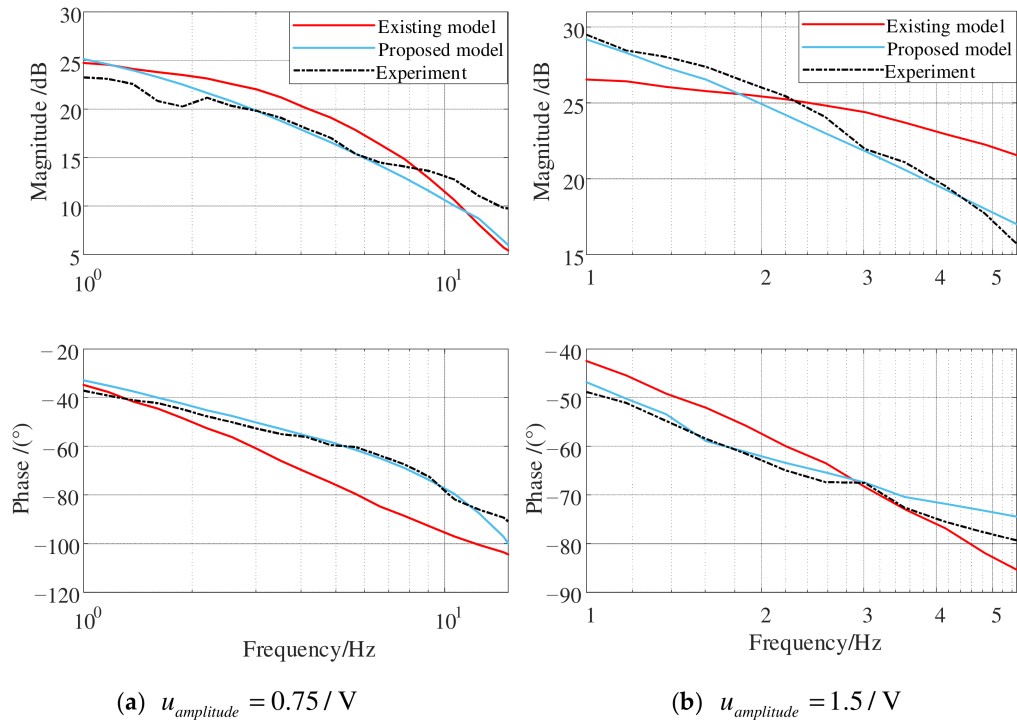

(**a**) $u_{amplitude} = 0.75 / V$        (**b**) $u_{amplitude} = 1.5 / V$

**Figure 17.** Simulated and experimental frequency responses under an open-loop.

### 4.3.3. Closed-Loop Time Response

After adding the closed-loop PI regulator, an additional set of experiments was conducted to test the ability of the model to reproduce the response of the system to setpoints. The comparison results shown in Figure 18a–d also reveal that the proposed model is much better than the existing model. The proposed model can accurately reproduce the zero-crossing dead zone and velocity fluctuation when following a sinusoidal signal, as well as the overshoot and oscillation times when following a square wave, while the existing model cannot. It can be seen from Figure 18 that the velocity of DMPTM fluctuates greatly after the closed-loop due to the difference in the clearance between the two transmission chains.

### 4.3.4. Closed-Loop Frequency Response

In the frequency domain, the closed-loop frequency responses of the model and the system at different speed command amplitudes are compared, and the results are shown in Figure 19a,b. The comparison results show that the proposed model accurately reproduces the system frequency response in terms of amplitude and phase. In particular, the 5.5 Hz resonance point after the system is closed is well captured, which shows that the backlash and friction nonlinearity of the system lead to a low bandwidth, which is not conducive to the high dynamic control of ISPs. However, the existing model cannot reproduce the closed-loop frequency domain characteristics of the system.

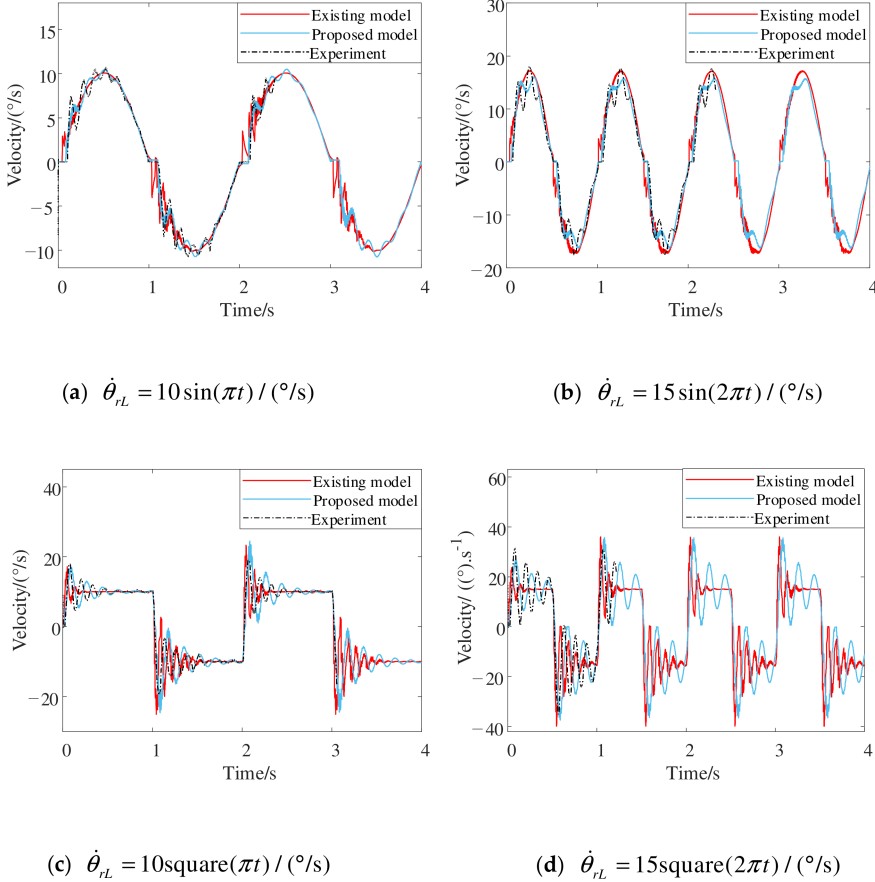

(**a**) $\dot{\theta}_{rL} = 10\sin(\pi t)\,/\,(°/s)$      (**b**) $\dot{\theta}_{rL} = 15\sin(2\pi t)\,/\,(°/s)$

(**c**) $\dot{\theta}_{rL} = 10\text{square}(\pi t)\,/\,(°/s)$      (**d**) $\dot{\theta}_{rL} = 15\text{square}(2\pi t)\,/\,(°/s)$

**Figure 18.** Simulated and experimental time responses under a closed-loop.

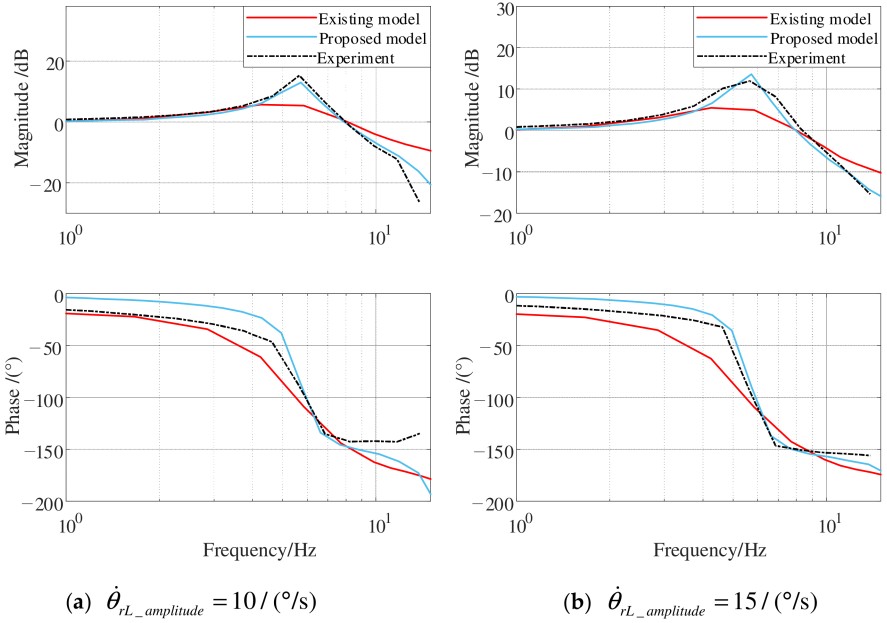

(**a**) $\dot{\theta}_{rL\_amplitude} = 10\,/\,(°/s)$      (**b**) $\dot{\theta}_{rL\_amplitude} = 15\,/\,(°/s)$

**Figure 19.** Simulated and experimental frequency responses under a closed-loop.

To quantify the accuracy of the model, the Pearson correlation coefficient was used to evaluate the fitting degree of the model response curve and the actual response curve. The Pearson correlation coefficients are calculated according to Figures 16 and 18, as shown in Tables 3 and 4. From the results, it can be seen that the fitting degree between the proposed

model and the actual is $r_{\text{open-loop}} > 99.41\%$ for the open-loop and $r_{\text{closed-loop}} > 83.7\%$ for the closed-loop, which verifies the accuracy of the model. Compared with the existing model, it can be improved by up to 5.61% and 1.78% in the open-loop and closed-loop, respectively.

**Table 3.** The Pearson correlation coefficient for open-loop time responses.

| Experiments | Existing Model/% | Proposed Model/% | Improvement/% |
|---|---|---|---|
| $0.6\sin(\pi t)/\text{V}$ | 94.13 | 99.41 | 5.61 |
| $0.75\sin(2\pi t)/\text{V}$ | 97.47 | 99.67 | 2.26 |
| $0.6\text{square}(0.8\pi t)/\text{V}$ | 99.14 | 99.68 | 0.54 |
| $0.75\text{square}(0.8\pi t)/\text{V}$ | 99.30 | 99.73 | 0.43 |

**Table 4.** The Pearson correlation coefficient for closed-loop time responses.

| Experiment | Existing Model/% | Proposed Model/% | Improvement/% |
|---|---|---|---|
| $10\sin(\pi t)/(°/\text{s})$ | 98.53 | 99.17 | 0.65 |
| $15\sin(2\pi t)/(°/\text{s})$ | 98.13 | 98.66 | 0.54 |
| $10\text{square}(\pi t)/(°/\text{s})$ | 92.77 | 94.42 | 1.78 |
| $15\text{square}(2\pi t)/(°/\text{s})$ | 82.75 | 83.70 | 1.15 |

## 5. Conclusions

High-performance control of ISPs can be achieved, provided that a reasonable knowledge of the dynamic model of the mechanical system is available. In this paper, the single components of the transmission chain are introduced in detail. Then, the overall linear dynamic model of the DMPTM is formed. In the nonlinear aspects of the system, the dead-zone model, considering the time-varying stiffness, is proposed to describe the system backlash, and the Stribeck model is used to analyze the friction of the system, which can more accurately depict the details of DMPTM in the process of speed reversal. The experimental results in the time domain show that the proposed model is highly consistent with the actual system. The fitting degree between the model and the actual speed response is $r_{\text{open-loop}} > 99.41\%$, for the open-loop, and $r_{\text{closed-loop}} > 83.7\%$, for the closed-loop, which are better than the results for the existing model. In the frequency domain, whether it is under the open-loop or the closed-loop, the fitting degree is very good, verifying the accuracy of the proposed model.

The research of this paper can provide a theoretic guide for the optimization of system performance and the design of high-precision controllers.

**Author Contributions:** Conceptualization, J.Z.; methodology, J.Z.; software, L.C.; validation, L.C., R.T. and B.L.; formal analysis, J.Z.; investigation, J.Z.; resources, J.Z.; data curation, J.Z.; writing—original draft preparation, J.Z.; writing—review and editing, D.F and X.X.; visualization, J.Z.; supervision, D.F and X.X.; project administration, D.F.; funding acquisition, D.F. All authors have read and agreed to the published version of the manuscript.

**Funding:** This research was funded by the National Key R&D Program of China (Grant No. 2019YFB2004700).

**Institutional Review Board Statement:** Not applicable.

**Informed Consent Statement:** Not applicable.

**Data Availability Statement:** Not applicable.

**Conflicts of Interest:** The authors declare no conflict of interest.

**Appendix A**

$$\overline{D} = \begin{bmatrix} \overline{d}_{1,1} & 0 & \overline{d}_{1,3} \\ 0 & \overline{d}_{2,2} & \overline{d}_{2,3} \\ \overline{d}_{3,1} & \overline{d}_{3,2} & \overline{d}_{3,3} \end{bmatrix}, \overline{K} = \begin{bmatrix} \overline{k}_{1,1} & 0 & \overline{k}_{1,3} \\ 0 & \overline{k}_{2,2} & \overline{k}_{2,3} \\ \overline{k}_{3,1} & \overline{k}_{3,2} & \overline{k}_{3,3} \end{bmatrix}$$

where

$$\overline{d}_{1,1} = D_3 + D_{34}R_3^2 C_{34}$$
$$\overline{d}_{1,3} = \overline{d}_{3,1} = -D_{34}R_3 R_4 C_{34}$$
$$\overline{d}_{2,2} = D_3 + D_{34}R_3^2 C_{34}$$
$$\overline{d}_{2,3} = \overline{d}_{3,2} = -D_{34}R_3 R_4 C_{34}$$
$$\overline{d}_{3,3} = D_L + 2D_{34}R_4^2 C_{34}$$
$$\overline{k}_{1,1} = K_{34}R_3^2 C_{34}$$
$$\overline{k}_{1,3} = \overline{k}_{3,1} = -K_{34}R_3 R_4 C_{34}$$
$$\overline{k}_{2,2} = K_{34}R_3^2 C_{34}$$
$$\overline{k}_{2,3} = \overline{k}_{3,2} = -K_{34}R_3 R_4 C_{34}$$
$$\overline{k}_{3,3} = 2K_{34}R_4^2 C_{34}$$
$$C_{12} = \cos^2 \alpha_1 \cos \delta_1$$
$$C_{34} = \cos^2 \alpha_3$$

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
