# Peer review of "An Elaborate Dynamic Model of the Dual-Motor Precision Transmission Mechanism for Performance Optimization"

_machines, doi:10.3390/machines10121181_

Round 1

Reviewer 1 Report

1. This paper lacks little in literature to justify the contribution of the work being presented

2. Need to present the proposed algorithm with step-by-step approach and flowchart to understand the operation

3. The effectiveness of the proposed dual-motor precision transmission mechanism (DMPTM) has not been validated compared with other relevant mechanism. Also kindly compare it with a recent technique that ensures the similar technical operations.

4. Ensure to enclose the parameters and nomenclature employed in the modular multi speed transmission system for clarification. The authors can elaborate the parametric details of level of accuracy obtained with suitable data.

5. The Model validation section lags with clarity, author supposed to explain in detail about the various loops with respect to time and frequency as well as the impact of DMPTM in the total process.

6. Give the comparative analysis of fitness function of proposed method with the other similar transmission mechanism.

Author Response

Thank you for your valuable suggestions to the manuscript, and I have made detailed modifications to the paper according to your comments.

Point 1: This paper lacks little in literature to justify the contribution of the work being presented.

Response 1: I have revised the introduction of this paper, added some new research results related to this paper, and highlighted the contributions of this paper.

Point 2: Need to present the proposed algorithm with step-by-step approach and flowchart to understand the operation.

Response 2: I have added Section 3.3 Overall dynamic model(It introduces the modeling process) in the paper to enable readers to have a more direct understanding of DMPTM modeling methods.

Point 3: The effectiveness of the proposed dual-motor precision transmission mechanism (DMPTM) has not been validated compared with other relevant mechanism. Also kindly compare it with a recent technique that ensures the similar technical operations.

Response 3: The application background of this paper is the weapon station, which has the characteristics of large volume and weight of its loads. In the process of scheme design, DMPTM was finally adopted to eliminate backlash after comparison and demonstration with other transmission structures and many scholars have used the DMPTM to eliminate backlash. Due to the length and focus of the research, this article did not give a detailed introduction to other relevant institutions. I am sorry for the trouble caused to you, and thank you for your suggestions.

Point 4: Ensure to enclose the parameters and nomenclature employed in the modular multi speed transmission system for clarification. The authors can elaborate the parametric details of level of accuracy obtained with suitable data.

Response 4: The method of obtaining model parameters has been introduced in more detail in Section 4 of this paper.

Point 5: The Model validation section lags with clarity, author supposed to explain in detail about the various loops with respect to time and frequency as well as the impact of DMPTM in the total process.

Response 5: The introduction of experimental items is added in the paper, and the analysis of the impact of DMPTM on system characteristics is added in the analysis of experimental results.

Point 6: Give the comparative analysis of fitness function of proposed method with the other similar transmission mechanism.

Response 6: Due to the project arrangement, the team has only machine one DMPTM, and lacks other forms of transmission mechanisms, so I am very sorry that the experiments you requested cannot be carried out.

Reviewer 2 Report

This paper proposes a dual-motor precision transmission mechanism as the drive system for inertially stabilized platforms, with the mechanism purpose being elimination of backlash.

A dynamic model of the DMPTM is formed. The results of the experiment conducted on the experimental rig are highly consistent with those of the proposed model, thus verifying its accuracy.  

However, the following issues have to be resolved:

In Chapter 1, while describing structure of the mechanism, authors mention L-shaped planetary reducer but on the Fig. 3, a simple bevel reducer is presented. Please explain!

In Chapter 2, eq.17, all members have to be defined – please define T! Why can damping be neglected in the vibration equation?

Chapter 3 as the key chapter in the paper has to be improved. Please define all members in eqs. 21 to 25. Development of the dynamical model (Figure 13.) has to be explained in more detail.  There are no references for Stribeck friction model, for analytical expressions as well as for the procedures for obtaining model parameters.

Since the concept of Dead zone model and Stribeck friction model are well known, please elaborate more precisely main contributions of this paper.

Author Response

Thank you for your valuable suggestions to the manuscript, and I have made detailed modifications to the manuscript according to your comments.

Point 1: In Chapter 1, while describing structure of the mechanism, authors mention L-shaped planetary reducer but on the Fig. 3, a simple bevel reducer is presented. Please explain!

Response 1: The L-type reducer is used in this paper. As shown in the figure below, the manufacturer named it L-type planetary reducer, but the main part is a pair of bevel gears. Therefore, in order to facilitate the analysis, this paper considers it as a simple bevel gear reducer.

Point 2: In Chapter 2, eq.17, all members have to be defined -please define T! Why can damping be neglected in the vibration equation?

Response 2: The definition of T has been added to the manuscript. The existence of damping does not affect the resonant frequency, but the resonant amplitude. Therefore, scholars usually calculate the undamped natural resonant frequency when calculating the natural resonant frequency of the system.

Point 3: Chapter 3 as the key chapter in the paper has to be improved. Please define all members in eqs.21 to 25. Development of the dynamical model (Figure 13.) has to be explained in more detail. There are no references for Stribeck friction model, for analytical expressions as well as for the procedures for obtaining model parameters.

Response 3: In the revised manuscript, all members in eqs 21 to 25 has been defined, and the explanation of Figure 13 was added. In addition, in the field of practical engineering application, the Stribeck model can reflect the characteristics of system friction, and has the advantages of simple structure, clear physical meaning, and easy parameter identification. Therefore, this paper uses the Stribeck friction model to describe the friction nonlinearity of the system, and supplements the relevant references. The acquisition process of model parameters is completed through offline testing and identification, and the method is conventional. As the current length of this paper has reached 18 pages, the process of parameter acquisition has not been explained in detail in this paper.

Point 4: Since the concept of Dead zone model and Stribeck friction model are well known, please elaborate more precisely main contributions of this paper.

Response 4: The main contributions of this paper have been introduced in more detail in the abstract, introduction and conclusion of the revised manuscript. The main contributions of this paper are as follows: (1) A case study of the complex transmission chain of the DMPTM is first worked out in detail, which deeply reveals the working principle of DMPTM and provides a model framework for the research of clearance elimination method and high-precision control method of the DMPTM; (2) The modified dead zone model takes into account the time-varying characteristics of meshing stiffness, which can more accurately describe the details of DMPTM in the process of speed reversal.

Round 2

Reviewer 2 Report

Point 1: In Chapter 1, while describing structure of the mechanism, authors mention L-shaped planetary reducer but on the Fig. 3, a simple bevel reducer is presented. Please explain!

Response 1: The L-type reducer is used in this paper. As shown in the figure below, the manufacturer named it L-type planetary reducer, but the main part is a pair of bevel gears. Therefore, in order to facilitate the analysis, this paper considers it as a simple bevel gear reducer.

FABR060-25-S2-P1 is L type planetary reducer which means that there is a planetary stage before bevel gear pair (between the motor and the bevel pair). So, rigidly concerning, presented model is not correct - whole module is missing. On the other side, this is high precision planetary gearbox (small backlash) and discussion before eq.16 is completely valid. I suggest to authors to add short explanation in chapter 2.3 about modeling LPR with bevel gear pair. It would be interesting to investigate, as the further research, the impact of addition of planetary stage to the friction model.

Please recheck the text, there are some minor problems - for instance "which causes the backlash characteristics cannot to be truly reflected" .

Author Response

Thank you for your valuable suggestions to the manuscript, and I have made detailed modifications to the manuscript according to your comments.

Point 1: FABR060-25-S2-P1 is L type planetary reducer which means that there is a planetary stage before bevel gear pair (between the motor and the bevel pair). So, rigidly concerning, presented model is not correct -whole module is missing. On the other side, this is high precision planetary gearbox (small backlash) and discussion before eq.16 is completely valid. I suggest to authors to add short explanation in chapter 2.3 about modeling LPR with bevel gear pair. It would be interesting to investigate, as the further research, the impact of addition of planetary stage to the friction model.

Response 1: According to the product manual, FABR060-25-S2-P1 is indeed a two-stage reducer. The first stage is planetary gear reduction, and the second stage is bevel gear reduction. Therefore, thank you very much for your suggestions. Since the planetary stage is the first stage, its backlash can be ignored, and its friction can be equivalent to the motor side. Therefore, the focus is on the bevel gear transmission. But I would add some explanation in chapter 2.3 about modeling LPR with bevel gear pair according to your suggestions.

Point 2: Please recheck the text, there are some minor problems - for instance "which causes the backlash characteristics cannot to be truly reflected"

Response 2: I would carefully recheck the full text and correct the errors.
